# Practical Models of Pharmaceutical Care for Improving Tuberculosis Patient Detection and Treatment Outcomes: A Systematic Scoping Review

**DOI:** 10.3390/tropicalmed8050287

**Published:** 2023-05-20

**Authors:** Ivan Surya Pradipta, Erya Oselva Yanuar, Chaca Yasinta Nurhijriah, Nadya Putri Maharani, Laila Subra, Dika Pramita Destiani, Ajeng Diantini

**Affiliations:** 1Departement of Pharmacology and Clinical Pharmacy, Faculty of Pharmacy, Universitas Padjadjaran, Bandung 45363, Indonesia; 2Drug Utilisation and Pharmacoepidemiology Research Group, Center of Excellence in Higher Education for Pharmaceutical Care Innovation, Universitas Padjadjaran, Bandung 45363, Indonesia; 3Faculty of Bioeconomic and Health Sciences, University Geomatika Malaysia, Kuala Lumpur 54200, Malaysia

**Keywords:** pharmacy, tuberculosis, community, hospital

## Abstract

Decreasing global tuberculosis (TB) notifications indicate problems related to TB patient detection and treatment outcomes. Pharmaceutical care (PC) has potential roles in managing these issues. However, PC practices have not yet become widespread in the real world. This systematic scoping review aimed to identify and analyze the current literature on practical models of pharmaceutical care for improving tuberculosis patient detection and treatment outcomes. We then discussed the present challenges and future considerations for the successful implementation of PC services in TB. A systematic scoping review was performed to identify the practice models of PC in TB. Systematic searches and screening were used to identify relevant articles in the PubMed and Cochrane databases. We then discussed the challenges and recommendations for successful implementation using a framework to improve professional healthcare practice. Our analysis included 14 of 201 eligible articles. We identified that the focuses in the PC of TB are on increasing patient detection (four articles) and improving TB treatment outcomes (ten articles). Practices cover services in the community and hospital settings, such as screening and referring people with presumptive TB, tuberculin test services, collaborative practices for treatment completion, directly observed treatment, the solution of drug-related problems, reporting and managing adverse drug reactions, and medication adherence programs. Although PC services positively increase TB patient detection and treatment outcomes, hidden challenges in the actual practice are analyzed. Several factors should be comprehensively considered in successful implementation, such as guidelines, individual pharmacy personnel, patient, professional interaction, organizational capacity, regulation, incentive, and resource factors. Hence, a collaborative PC program that involves all related stakeholders should be considered to create successful and sustainable PC services in TB.

## 1. Introduction

Tuberculosis (TB) is a seriously threatening disease worldwide. A global report estimated that 10.6 million people contracted TB, and 1.4 million died in 2021 [1]. Multidrug-resistant TB (MDR-TB), an evolved pathogen that is resistant to two powerful anti-TB drugs (isoniazid and rifampicin), poses a major problem. The global treatment success rate of MDR-TB remains low, at 60% [1]. The spread of COVID-19 during the current pandemic and its global spread complicates the picture further. New notification TB cases worldwide significantly dropped from 7.1 million (2019) to 5.8 million (2020) [1], indicating a number of potentially missed TB cases and poor treatment outcomes, leading to increased TB incidence and mortality in the coming years.

The updated global TB report described that the complexity of TB-related problems in countries with high TB prevalence leads to the largest contribution in decreasing global TB notification [1]. It highlights the potential problems of TB patient detection and treatment. Previous studies have demonstrated that restricted patient access to diagnosis and treatment and limited healthcare staff availability at community health centers are barriers to successful TB treatment in high-prevalence TB countries [2,3]. These works report that the current healthcare systems are unable to detect, treat, and report TB cases optimally. An improved engagement program for all resources is essentially needed to overcome problems related to TB patient detection and treatment outcomes.

Pharmacy personnel play a role in managing TB. Patient pathway studies show that pharmacies are primary facilities for first aid for people affected by TB [4,5] and can be a potential site for increased TB case detection. TB treatment complexity is greater in hospital environments due to various types of critically ill patients. Pharmacotherapeutic follow-up is essential for these patients since they may have more potential drug-related problems (DRPs), especially when they have other associated diseases in addition to TB [6].

A joint statement by the International Pharmaceutical Federation (FIP) and World Health Organization (WHO) urged national TB programs and national pharmacy associations to jointly implement TB programs [7]. However, this has not seen widespread practical implementation [6]. Because improving the practice of health professionals involves complex determinants [8], studies of practice models and factors of their successful implementation are needed. This will provide insight for developing modern pharmaceutical care (PC) to improve tuberculosis patient outcomes and minimize healthcare costs, especially in high-burden TB countries. Therefore, this study aimed to review various models of pharmaceutical services and their effects on improving TB patient detection and treatment outcomes. We then analyzed challenges and future considerations for the successful implementation of PC services in TB.

## 2. Materials and Methods

### 2.1. Study Design and Search Strategy

A systematic scoping review was performed to identify models of PC practice in TB. We then analyzed challenges and developed recommendations for successful implementation, observing the framework for improvements in professional healthcare practices developed by Fottorp et al., 2013 [8]. We followed the Preferred Reporting Items for Systematic Reviews and Meta-Analyses guidance for the transparent and systematic report [9].

A study protocol was developed prior to the study. Since a scoping review protocol cannot be registered in the PROSPERO (a prospective international registration for systematic reviews), the unregistered protocol was internally used as a study guide for the research team. According to the study protocol developed, a comprehensive search strategy was used to retrieve relevant articles from two reputable medical databases. PubMed, a medical research database, was searched to identify relevant articles, and Cochrane Library, indexed articles in area intervention studies, was used to ensure a comprehensive search. Pharmaceutical services and TB terms combined medical subheadings (Mesh), and Boolean operators were also used. The search was conducted on texts published on 31 July 2022, without language or publication date restriction. The search strategies were developed involving several key terms, i.e., “Pharmacy”, “Pharmacies”, “Pharmacy Service”, “Hospital”, “Pharmacy Technicians”, “Community Pharmacy Services”, “Pharmacy Research”, “Pharmaceutical Services”, “Tuberculosis”, “Latent Tuberculosis”, “Multidrug-Resistant”. The full search strategies are given in Appendix A.

### 2.2. Study Criteria

We defined the PECOS (population, exposure, comparison, outcomes, and study type) to obtain the relevant articles in this review. The population was presumptive, latent, and active tuberculosis subjects, while the exposure was PC intervention in community and hospital settings. The study comparator was a care without PC interventions or descriptive situations before the PC was applied in pre–post interventional studies. Improving TB case detection and treatment outcomes were defined as the study outcome in this review.

In terms of study type, we included original articles that analyzed the effects of PC interventions on TB. We excluded review articles, case reports, case series, commentaries/editorials, book chapters, study protocols, guidelines, and articles that only assessed baseline information for implementing PC.

### 2.3. Article Selection, Data Extraction, and Analysis

Selected articles from the databases were reviewed using the study criteria in two steps. The first step focused on the title and abstract, followed by full-text reviews in the second step. Three reviewers (E.O.Y., C.Y.N., and N.P.M.) independently performed title–abstract and full-text screening. Discrepancies were solved in intensive discussions with the fourth reviewer (I.S.P.). Since we conducted a scoping review that provided an overview of the existing evidence regardless of methodological quality, risk of bias assessments was not performed in our study [10].

In the full-text review, we assessed article eligibility in detail regarding their population, intervention, comparator, outcome, and design. Essential information was extracted for articles fitting the criteria, i.e., the authors, objective, design, study period, location, target population, PC models, and outcomes. The effect measures for study outcome were assessed for each study. The effect measurements of the study outcomes were descriptive (percentage, number) or comparative (odds ratio or relative risk) analyses. Three reviewers (E.O.Y., C.Y.N., and N.P.M.) conducted the initial data extraction, and the fourth reviewer (I.S.P.) made the final decision. A qualitative synthesis was performed to map pharmaceutical models across studies, and the practical challenges and recommendations were described in the narrative analysis.

## 3. Results

During the search, we identified 199 records from the PubMed database and ten records from the Cochrane Library database. We found eight duplicate records. A total of 201 articles were screened for the title and abstract. This initial screening excluded 187 irrelevant articles with various reasons, i.e., incompatible intervention (108), outcome (1), population (18), and type of articles (i.e., case report/series (4), review (45), book chapter/report (2), protocol (3), commentary/editorial (5), and guideline (1)). We finally performed a full-text screening of 14 articles for the qualitative analysis. The heterogeneity in populations, interventions, and outcomes across the included studies prevented quantitative analysis. The flow diagram of the included articles is presented in Figure 1.

We obtained studies from high- and low-prevalence tuberculosis countries, namely Pakistan [11], Vietnam [12], Bolivia [13], the United States [14,15,16,17,18], Thailand [19], Spain [20], China [21], Turkey [22], Indonesia [23], and Brazil [24]. PC interventions for latent TB infection (LTBI) and presumptive and active TB populations were studied. PC services were conducted in hospital (five studies) and community (nine studies) settings. We identified two main outcome orientations for PC intervention, i.e., increasing TB patient detection and improving TB treatment outcomes. Characteristics of the included studies are presented in Table 1.

### 3.1. PC in Increasing TB Patient Detection

Five studies presented means of increasing TB patient detection. Three focused on how pharmacy personnel can screen presumptive TB patients and refer them to appropriate facilities for further TB examination and diagnosis [11,12,13], and two focused on pharmacists’ ability to administer tuberculin skin tests (TSTs) to support LTBI diagnosis [15,17]. PC models and their effects on increasing TB patient detection are presented in Table 2.

In screening and referring activities, pharmacy personnel identified presumptive TB patients among customers at their pharmacies. Pharmacy personnel directly communicated with customers in exploring TB signs and symptoms, such as persistent cough, weight loss, night sweats, high temperature, fatigue, loss of appetite, and neck swelling. Once a presumptive TB patient was identified, pharmacy personnel recommended further TB examination at a particular health facility.

The number of pharmacies included in these studies ranged between 70 and 100. The ability of pharmacy personnel to refer presumptive TB patients depended on the number of pharmacies engaged. A Pakistani study successfully referred 3025 visitors to health facilities from 500 pharmacies [11], while studies in Vietnam and Bolivia only referred 310 and 41 individuals to health facilities from 150 and 70 pharmacies, respectively [12,13]. However, the data show that not all visitors followed pharmacy recommendations. We noted that the number of visitors following the pharmacy’s recommendation ranged from 11 to 1901. Furthermore, studies showed that a few visitors were then diagnosed with TB. Studies from Pakistan, Vietnam, and Bolivia showed successful diagnoses of TB in 547, 10, and 3 individuals, respectively.

Two studies from the US indicated that pharmacists were granted the authority to prescribe, administer, and read the TSTs [15,17]. This provides a wide access to TSTs, which can help support LTBI/TB diagnosis. Pharmacists should follow their training to maintain their TST license. Two studies reported that 43 and 2 pharmacists participated in each program; the total number of those tested in the two studies was 578 and 18, respectively. However, not all of those examined returned for the test reading. We noted that in those two studies that 536 and 17 people revisited the pharmacy for the reading test, with positivity rates of 3.1% and 0%, respectively. Most patients in studies with a zero positivity rate were from a low-risk LTBI/TB population, and the tests were generally needed for job requirements [17]. The average time for conducting the test in the pharmacy was less than 10 min [17].

### 3.2. PC in Improving TB Treatment Outcomes

We identified nine studies with outcome orientations of improving TB treatment outcomes [14,18,19,20,21,22,23,24]. Of these, four focused on community pharmacy services and five on hospital services. Those focused on community pharmacy included three which specifically analyzed the role of pharmacy in improving LTBI treatment [14,16,18], and one investigated community pharmacists as directly observed treatment (DOT) providers to improve the rational use of TB medication [20]. In the hospital pharmacy service, all studies analyzed PC activities in the hospital regarding the assessment, monitoring, and follow-up of patient DRPs, educational programs for medication adherence, direct/indirect scheduled meeting, and medication consultation [19,21,22,23,24]. PC models and their effects on TB treatment outcomes are presented in Table 3.

In LTBI care, collaborative practices between community pharmacists and healthcare staff are described in three US studies. These studies highlighted the community pharmacist’s role in supporting treatment completion in LTBI patients. One study focused on a refugee population [16], while the others examined the general LTBI population [14,18]. Generally, community pharmacists conducted regular meetings for assessing and reporting DRPs (including adverse drug reactions), drug monitoring, and counselling to support treatment completion in LTBI patients. Contributing community pharmacists produced a relatively high treatment completion of 59–91%. Most incomplete treatments were due to ADR, and the remainder were due to a lack of perception of treatment benefits or moving to another health facility. A study found that collaborative LTBI care between community pharmacists and public health departments reduced as much as 143 h of labor for the public health department [14].

In active TB care, one study demonstrated a community pharmacy providing DOT services to improve medication adherence in high-risk non-adherent populations [20]. Collaborators included the pharmacist, a hospital pulmonologist, and a part-time social worker. The pharmacist provided several services during each patient visit, including reinforcing the importance of treatment adherence, inquiring about DRP events, reminding the patient of upcoming appointments at the pharmacy and hospital, and offering required sociosanitary support. That study reported that the community pharmacists successfully identified 108 DRPs. In a comparison between a PC intervention group and a self-administrated treatment (SAT) group, treatment completion was significantly higher in the PC intervention group (RR 3.07; 95%CI 2.13–4.41), and failed treatment was less likely in the PC group (RR 0.33; 95%CI 0.22–0.50) [20].

In hospital settings, we found that hospital pharmacists contributed to the improvement of TB treatment outcomes through several services, including assessing, monitoring, following up, and reporting potential and actual DRPs, lab checks for identifying ADR, providing face-to-face or phone drug consultation, and administering educational programs for medication adherence with standardized written and oral counseling. An observational study reported that in a Thai hospital, PC intervention had the highest proportion of treatment success (94.90%; CI 91.57–95.63), better than home visits (93.60%; CI 91.57–95.63) and modified DOT (90.10%; CI 87.54–92.66) [19]. However, another study from China found no significant difference between PC intervention and usual care for TB treatment outcomes (i.e., treatment success, failure, default, transfer out, death, or sputum conversion time) [21].

Furthermore, two studies have shown that the attendance of patient visits was significantly higher in the PC intervention group than in the PC intervention (*p* = 0.018 and *p* ≤ 0.005) [21,22]. Pharmacist involvement has driven DRP findings among patients with TB in hospitals [21,22,24]. A range of 28–128 DRPs were identified. These consisted of inappropriate medication, additional medications needed, inappropriate doses, adverse drug reactions, and non-adherence to medication. Most DRPs were resolved by pharmacists. Two studies showed the benefits of pharmacist counselling in enhancing medication adherence [22,23]. Standardized counselling led by the pharmacist was significantly associated with patient attendance at the hospital, compared with the non-pharmacist intervention [22], while a combination of pharmacist counseling and written material (leaflet) in improving medication adherence was superior to the other two groups, i.e., a group that only receives counseling and a group without counseling and leaflets [23].

## 4. Discussion

Our review found that PC models focused on two outcomes, i.e., increasing TB patient detection and improving TB treatment outcomes. For increased TB patient detection, two practice models in the community setting screened the presumptive TB patients and referred them to the health facility for further diagnosis; TSTs are provided in community pharmacies for the general population to improve public accessibility and examine the status of LTBI. Concerning TB treatment outcomes, we identified PC practice in the community and hospital settings. In the community setting, PC activities described collaborative practices as improving LTBI medication completion rates and participation in the DOT program for improving medication adherence in a high-risk group of non-adherence to medication. In hospital settings, PC activities focused on improving TB treatment outcomes through several activities of the hospital pharmacist, such as assessing, monitoring, following up on potential and actual DRPs, and reporting on DRPs, including ADR. The other activities conducted in the hospital pharmacy are face-to-face or telephone drug consultations and educational programs for medication adherence with standardized written and oral counseling. Most PC services positively impact treatment success. However, another study showed no significant effect of PC on improving treatment success, highlighting various effects and hidden challenges in actual practice.

We identified different forms of care across the PC practice scenario. Prevention and timely treatment actions are more frequent in the community setting than in the hospital setting, while the management of serious patient conditions is part of pharmacy personnel’s actions to improve TB treatment outcomes in the hospital setting. These different forms of care underline the need for different approaches to engaging pharmacy personnel in TB care based on their practice setting. In the internal aspect, comprehensive knowledge and skill in managing severe TB conditions, patients with complications, and clinical data interpretation are mainly needed for hospital pharmacists in TB care [25]. Those are essential to developing a pharmaceutical care plan that can fully support the rational use of medicine by identifying and solving potential and actual DRPs among the complex problem of hospitalized patients with TB [26]. In the community setting, pharmacy personnel should have epidemiological and public health knowledge besides pharmacotherapy and pharmacovigilance knowledge. Importantly, the accessibility of pharmacy personnel to lab tests for identifying the potential ADR of TB medications is required in the community setting. It will support pharmacy personnel in managing medication adherence among patients with TB. However, the effects of PC practice are influenced by local contexts, involving several factors, such as individual pharmacy personnel, patient awareness, clear guidelines, professional interaction, incentives, resources, capacity for organizational change, and social, political, and legal factors [8].

Among the individual aspects, the positive awareness of the importance of TB services from pharmacy personnel and the availability of guidance are essential. The willingness of pharmacy personnel to join a TB program has been reported as a challenge in the activities of TB patient detection [11]. Studies in Vietnam and Pakistan showed that personnel regarded it as lost pharmacy income if they referred a presumptive TB patient to a health facility [11,12]. Pharmacy personnel should have adequate knowledge and skill in managing TB in their facility [11,15]. A multicross-sectional study reported that TB training is associated with TB patient detection among community pharmacy personnel in Indonesia [27]. Previous studies showed that a lack of exposure to TB training might lead to a limited understanding of how to manage presumptive or active TB patients [15,24]. Challenges will arise if pharmacy personnel is not accompanied by guidance in TB practices. This guidance could be a supporting document providing standardized practices for managing TB patients [8]. Hence, TB awareness, knowledge, skills, and guidance are essential for the success of pharmacy personnel in managing TB.

We found that a lack of incentives, high workload due to limited staff [27,28], and an unavailability of an adequate facility (e.g., consultation room, tracing system, record system for DRPs, and transfer outpatient) were reported as challenges [14,15,20,22,24]. Regarding new initiatives for PC practice, healthcare providers commonly consider the benefit and risks of providing the practices. Individual incentives are a consideration. An analysis of 19 case detection studies indicated that incentives should be considered in the recruitment of pharmacy personnel in TB patient detection programs [28]. The care taken for pharmacy income may worsen when profit is the primary goal of the pharmacy owner [28]. This can affect the capacity for organizational change to support TB.

In addition, workload, limited proper facilities, and drug availability were associated with limited patient case detection and improved treatment outcomes in TB care. Studies found that high workload was associated with TB patient detection in community pharmacies [27]. Moreover, a study in a hospital setting demonstrated that the limited activities performed by pharmacy personnel in managing TB patients were due to personnel limitations [24]. In terms of the facility, the availability of a consultation room for TB counselling, a tracing system for non-adherent/transfer-out patients, and integrating DRP reports with the medical record were beneficial for supporting PC in TB [14,22,24]. Another challenge was the absence of an integrated system between pharmaceutical services with TB stakeholders and local/national TB programs. The lack of connectivity among pharmacies, laboratories, and NTPs was a barrier to successful TB patient detection programs in Bolivian community pharmacies [13]. The lack of preparation of related facilities for TB patient detection in the pharmacy programs (e.g., laboratories for sputum tests, clinics, and primary care) is a primary factor for the low positivity rate of presumptive TB patients. Many presumptive TB patients referred by pharmacy personnel were reluctant to visit laboratories, clinics, or primary health facilities due to long queues, limited facilities for sputum tests, and complex procedures required for medical examination [11]. In the hospital setting, professional interaction among TB healthcare teams should be intensified to follow up on DRPs and non-adherent patients [24].

Following the principle of patient-centeredness, we identified several barriers in the community and hospital settings from the patient’s perspective. In TB patient detection programs at community pharmacies, services that patients should pay for out-of-pocket after receiving advice from pharmacy personnel for further TB examination may be a reason for not pursuing further TB examination [11,12]. Moreover, negative public perceptions of primary public health services were reported, which prevented presumptive TB patients from going to public health facilities [12]. This includes complex bureaucracy, long wait times, poor quality labs, and accessibility and privacy issues [12]. The lack of information on TB detection programs in community pharmacies affected the awareness of their recommendations among pharmacy personnel [13]. Furthermore, patient access to a health facility, worry about the potential drug side effects, adverse drug reactions, and complex and lengthy treatment regimens prevented successful pharmaceutical services [14,16,22].

Several limitations of this review should be acknowledged for a cautious interpretation. First, other PC practice models may be found in other databases or unpublished articles. Second, the heterogeneity of the population, intervention, comparison, and outcomes across the studies examined may lead to difficulties in performing a meta-analysis. Third, potential bias may exist in the included studies. Fourth, there is no economic impact of the PC service in our included study. To ameliorate this, we searched two reputable databases without language or study period restrictions. We also used an independent team to select and screen articles based on the study criteria to minimize potential information bias in our study.

Innovative strategies that engage all resources should be adopted to stabilize TB management in the COVID-19 era, especially in countries with a high prevalence of TB. Our review demonstrated that pharmaceutical services could improve TB case detection and treatment outcomes. However, several pre-conditions should be considered for consistent and satisfactory effects of PC services. First, an integrated program involving not only pharmacy personnel as the central actor in the PC services, but also national TB programs, central/local government, professional organizations, and relevant stakeholders should be initiated. The integrated program should describe the professional interactions between pharmacy personnel and TB stakeholders, supported by the comprehensive practical guidelines for providing TB services. Regulations should be developed by local leaders, TB programmers, and professional organizations to build friendly working environments for program implementation. Program monitoring and intensive communication regarding barriers, conflicts of interest, and TB awareness can be led by local leaders or TB programmers. Second, attractive TB training, including structured programs, incentives, and certificates, can be developed for pharmacy personnel to attract and improve their awareness, knowledge, and skills in managing TB. The specific license can be given to pharmacy personnel, following comprehensive training in the particular TB practice. Professional organizations and academic institutions can develop the program as part of a continuing educational program for pharmacy personnel. Third, standardized, accessible facilities for diagnosis and treatment, such as pharmacies, laboratories, primary care, and referral hospitals, are critical for supporting the pharmaceutical program in improving TB detection and treatment outcomes. Fourth, program socialization regarding TB pharmaceutical services may enhance public awareness of the pharmacy’s role and benefits, and in particular, pharmaceutical services in TB. Fifth, digital technology can be developed to reduce a pharmacy personnel’s workload in managing TB patients [29] and supporting activities related to TB recording and reporting. Finally, further studies on the clinical and economic impact of PC services are comprehensively needed.

## 5. Conclusions

Our review has identified PC models in increasing TB patient detection and improving treatment outcomes. The PC practices cover the area of community and hospital settings with several types of practice models, such as screening and referring people with presumptive TB, providing TSTs, collaborative practices for treatment completion, directly observed treatment, addressing DRPs, and reporting and managing adverse drug reactions and medication adherence programs.

This review indicates that the current pharmaceutical services could be beneficial for supporting TB control. Several factors should be comprehensively considered by pharmacy personnel and relevant stakeholders for successful implementation, such as: guidelines availability; sufficient individual knowledge and skill of pharmacy personnel; patient awareness for the pharmacy personnel roles; good professional interaction, incentives, and resources availability; strong capacity for organizational change; and existing regulation to support PC services. Hence, an integrative PC program for TB patient care should be considered for a successful program involving all related PC service stakeholders. Study on further implementation is needed to analyze the clinical and economic impacts of pharmaceutical services, considering the aforementioned comprehensive factors.

## Figures and Tables

**Figure 1 tropicalmed-08-00287-f001:**
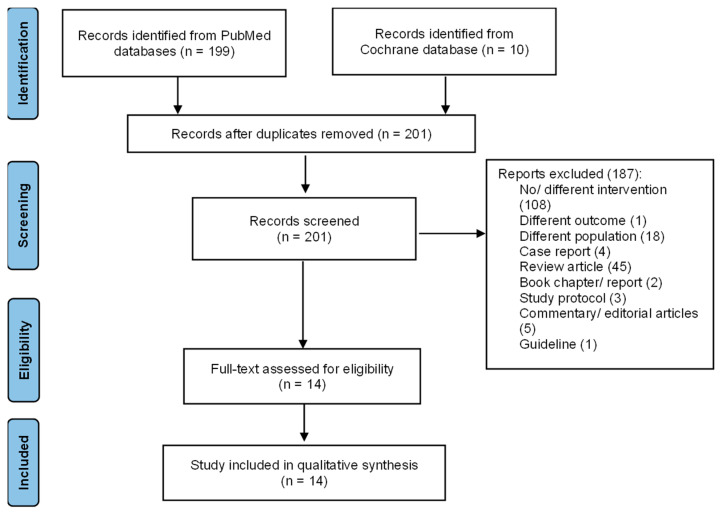
Flow diagram of the included articles.

**Table 1 tropicalmed-08-00287-t001:** Characteristics of the included studies.

No	Authors, Year	Objective	Design	Period	Location	Targeted Service	Outcome Orientation
1	Ullah et al., 2020 [11]	Assessing the effectiveness of TB case detection activities by pharmacy personnel.	Program implementation	January to December 2017	Three areas in Pakistan	Presumptive TB patients	Increasing TB patient detection.
2	Lonnroth et al., 2003 [12]	Describing community pharmacist attitude in the program of TB referral system for presumptive TB patients and evaluating the program’s feasibility.	Mixed methods followed the implementation program	April 2001 to December 2002	Ho Chi Minh, Vietnam	Presumptive TB patients	Increasing TB patient detection.
3	Lambert et al., 2005 [13]	Providing interventions to community pharmacists in decreasing anti-TB sales in pharmacies and increasing pharmacy personnel activities on the referral of presumptive TB patient activities to the health facility.	Intervention study	January to March 2002	Cochabamba, Bolivia	Presumptive TB patients	Increasing TB patient detection.
4	Jakeman et al., 2015 [15]	Describing community pharmacy experiences in TST program.	Program implementation	March 2011 to August 2013.	New Mexico, USA	Risk population of LTBI patients	Increasing TB patient detection.
5	Jakeman et al., 2020 [14]	Evaluating collaborative TB care between the public health department and community pharmacy in managing newly LTBI patients.	Program implementation	February 2017 to April 2018	New Mexico, USA	Newly diagnosed LTBI patients	Improving TB treatment outcomes.
6	Tanvejsil et al., 2016 [19]	Comparing TB treatment success among three services in TB patients, i.e., PC, home visit, and modified DOT.	Observational study	February 2017	Songkhla, Thailand	Adult pulmonary TB patients.	Improving TB treatment outcomes.
7	Juan et al., 2006 [20]	Comparing DOT through pharmacies with SAT in high-risk patients of non-adherence.	Observational study	1 January 1999 to 31 December 2002	Valencia, Spain	High risk of non-adherent TB patients *	Improving TB treatment outcomes.
8	Carter et al., 2017 [16]	Assessing the effect of a clinical pharmacist-run clinic for LTBI on the treatment completion rates in refugee patients.	Observational study	August 2012 to April 2016	Philadelphia and Pennsylvania, USA	Refugees with LTBI	Improving TB treatment outcomes.
9	Tang et al., 2018 [21]	Assessing the effect of PC on TB treatment outcomes compared with the UC.	A randomized case–control study	June 2018	Shanghai, China	Pulmonary TB patients who have already received a TB regimen of less than one month	Improving TB treatment outcomes.
10.	Clark et al., 2007 [22]	Assessing the effect of educational programs provided by pharmacists to improve medication adherence compared with a group without the educational program.	A randomized case–control study	August 2001 to February 2002	Istanbul, Turkey	Newly diagnosed TB and MDR-TB patients	Improving TB treatment outcomes.
11	Karuniawati et al., 2019 [23]	Assessing the effect of pharmacist counselling using written information (leaflet).	Quasi experimental	July to December 2017	Surakarta, Indonesia	Adult TB patients who have received TB medication for at least one month	Improving TB treatment outcomes.
12	Lopes et al., 2017 [24]	Describing and evaluating the effect of PC on TB patients.	Observational study	August 2009 to July 2012	Belo horizinte, Brazil	Adult TB outpatient	Improving TB treatment outcomes.
13	Hess et al., 2009 [18]	Describing treatment completion rates for LTBI in a community pharmacy in a college campus.	Observational study	2000–2006	California, USA	University students diagnosed with LTBI	Improving TB treatment outcomes.
14	Hecox, 2008 [17]	To report experiences of pharmacist-provided TSTs.	Observational study	February to December 2006	Washington, USA	Public who will have TSTs	Increasing TB case detection.

Information: * High risk of non-adherent TB patients: human immunodeficiency virus (HIV) infection, alcoholism, illicit drug use, immigrant or homeless status and/or previous failure to complete; TB: tuberculosis; TST: tuberculin skin test; PC: pharmaceutical care; LTBI: latent tuberculosis infection; USA: the United States of America; SAT: self-administrated treatment; DOT: directly observed treatment; UC: usual care; MDR-TB: multidrug-resistant tuberculosis.

**Table 2 tropicalmed-08-00287-t002:** Pharmaceutical care models and the effects on increasing tuberculosis patient detection.

No.	PC Models	Authors	Setting Area	Outcomes
1	Pharmacy personnel screened TB signs and symptoms in their pharmacy to identify presumptive TB patients, then referred the patient to a health facility for further TB examination.	Ullah et al., 2020 [11]	Community	The community pharmacy engaged: 500 pharmacies; The visitors referred by the pharmacy: 3025 visitors; The visitors who have visited HCF after the referral was made: 1901 visitors; The visitors diagnosed with TB: 547 visitors.
		Lonnroth et al., 2003 [12]	Community	The community pharmacy engaged: 150 pharmacies; The visitors referred by the pharmacy: 310 visitors; The visitors who visited HCF after the referral was made: 149 visitors; The visitors diagnosed with TB: 10 visitors.
		Lambert et al., 2005 [13]	Community	The community pharmacy engaged: 70 pharmacies; The visitors referred by the pharmacy: 41 visitors; The visitors who visited HCF after the referral was made: 11 visitors; The visitors diagnosed with TB: 3 visitors.
2	Pharmacists were granted the authority to prescribe, administer, and read TSTs.	Jakeman et al., 2015 [15]	Community	The total number of pharmacists included in the program: 43 people; The total number of patients tested: 578 people; The number of patients returned for the test interpretation: 536 people; The positivity rate: 3.1%.
		Hecox, 2008 [17]	Community	The total number of pharmacists included in the program: 2 pharmacists; The number TSTs administered: 18 TSTs; The average duration of TSTs: less than 10 min; The number of patients returned for the test interpretation: 17 patients; The positivity rate: 0%.

Information: TB: tuberculosis; TST: tuberculin skin test; PC: pharmaceutical care; HCF: healthcare facility.

**Table 3 tropicalmed-08-00287-t003:** Pharmaceutical care models and the effects on improving tuberculosis treatment outcomes.

No.	PC Models	Authors	Setting Area	Outcomes
1.	The community pharmacist provided weekly DOT services for newly diagnosed LTBI patients for 12 weeks. The PC included evaluating drug interaction and drug toxicity, dispensing LTBI regimen, monitoring medication adherence, and reporting the incidence of ADR to the relevant public health department.	Jakeman et al., 2020 [14]	Community	Total participants included: 40 patients; Total visiting patients: 398 patients; Patients with the completed treatment: 75%; Patients with uncompleted treatment due to ADR: 17.5%; Patients with uncompleted treatment due to loss of follow-up: 7.5%; Total patients with ADR: 60%; Patients with serious ADR reported to health department: 7 patients; Total saving time of the health department in the collaborative program: 143 h.
2	After the LTBI patients were diagnosed by a student health clinician and referred to the pharmacy, the pharmacist provided scheduled counseling with the patient on the importance of treating LTBI; medication adherence assessment by counting the pill or patient self-reported; assessment of ADR and risk factors of failed treatment.	Hess et al., 2009 [18]	Community	The number of patients analyzed: 348 patients; Six-month completion rate: 67%; Nine-month completion rate: 59%; The number of patients who withdrew: 41%; The reason for withdrawal: lack of perception of treatment benefit (52.50%) and discharging back to the primary care provider due to non-compliance with the schedule (41.20%); The patients who did not complete the treatment were more likely to have a rash and fatigue.
3.	Pharmacists conducted a monthly visit to the refugee diagnosed with LTBI with several activities, i.e., assessing drug-related problems (e.g., drug interaction, ADR), following up on the DRPs occurred (lab examination and drug modification), and patient counseling for medication adherence.	Carter et al., 2017 [16]	Community	The total refugees screened: 436 people; The refugees diagnosed with LTBI: 121 people; The patients included in the clinic program: 103 patients; The patients who completed treatment in the program: 94%; The patients who needed pharmaceutical intervention regarding medication adherence: 40%.
4.	The community pharmacists included in the district-based DOT program comprised a hospital pulmonologist staff, the community pharmacist, and a part-time social worker. After the high-risk criteria of TB patients were diagnosed and discharged from the hospital, the patients were assigned to continue the medication at the nearest community pharmacist where they lived. The community pharmacist provided several services during each patient visit, such as reinforcing the importance of treatment adherence, making inquiries about the occurrence of any drug-related event, reminding the patient about forthcoming appointments at the pharmacy and hospital, and offering the required sociosanitary support. The pharmacist initiated contact by phone or home visits to identify the reasons for default and to ensure the continuation of treatment.	Juan et al., 2006 [20]	Community	Total patients included in the group of DOT through pharmacy: 101 patients; Total patients included in the group SAT: 112 patients; Total DRPs reported in the DOT group: 108 DRPs; Total DRPs reported in the SAT group: 42 DRPs; The completed (RR 3.07; 95%CI 2.13–4.41) and failed treatment (RR 0.33; 95%CI 0.22–0.50) was a significant difference between DOT and SAT groups; No significant difference between the two groups regarding death due to HIV, transferred patients, bacteriological cure, admission to hospital, and relapse; The implementation of DOT increased the treatment cost by EUR 400 per patient.
5.	In the PC service, the clinical pharmacist provides TB care when the patient visits the hospital, including educational intervention, assessment of drug-related problems, drug monitoring (ADR and medication adherence), and phone consultation for drug-related issues.	Tanvejsilp et al., 2016 [19]	Hospital	The percentage of treatment success in the PC intervention was the highest (94,90%; 95%CI 91.57–95.63). It was followed by the home visit (93.60%; 95%CI 91.57–95.63) and the modified DOT (90.10%; 95%CI 87.54–92.66).
6.	Clinical pharmacists provided scheduled face-to-face or phone communication with TB patients regarding the educational program for medication adherence and successful treatment, identification DRPs, drug monitoring for ADR (e.g., lab check for ADR), and its follow-up.	Tang et al., 2018 [21]	Hospital	No significant differences between the two groups related to treatment outcomes (treatment success, failure, transfer out, default, death, and sputum conversion time); The significant differences between the PC and UC groups were found in the attendance of all scheduled visits (81% vs. 60%, *p* = 0.018); The number of DRPS identified and intervened: 57 DRPs; The number of DRPs solved: 50 DRPs.
7.	Pharmacists provided standard written and oral TB education for the TB inpatients and provided outpatient TB care with a routine schedule after they were discarded from the hospital. The PC also assessed the DRPs of TB patients.	Clark et al., 2007 [22]	Hospital	The attended visit was significantly greater in the group of PC than non-PC (*p* < 0.005); The DRPs were identified: 28 issues; The DRPs were solved by clinical pharmacist intervention: 21 issues.
8.	The hospital pharmacist provided oral and standardized written TB counseling to improve medication adherence.	Karuniawati et al., 2019 [23]	Hospital	The proportion of patients who received counseling and leaflets was the highest in improving medication adherence. It was followed by the group of patients who only received counseling and control (without counseling and leaflet) groups.
9.	The hospital pharmacist provided routine consultation, performed pharmacology and non-pharmacology intervention, and assessed DRPs and ADR with the standardized instrument.	Lopes et al., 2017 [24]	Hospital	The number of patients assessed: 62 patients; The number of DRPs identified: 128; The number of interventions performed by the pharmacist: 115 intervention; The DRPs were identified to be: unnecessary drug therapy, additional therapy, low dosage, adverse drug reaction, and non-adherence); The outcome of the intervention: positive (68.70%), negative (17.40%), and not evaluated (13.9%).

Information: TB: tuberculosis; PC: pharmaceutical care; LTBI: latent tuberculosis infection; DOT: directly observed treatment; UC: usual care; ADR: adverse drug reaction; SAT: self-administered treatment; DRPs: drug-related problems; RR: relative risk; 95%CI: 95% confidence interval.

## Data Availability

The original contributions presented in the study are included in the article/Appendix A. Further inquiries can be directed to the corresponding author.

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
