# Peer review of "Practical Models of Pharmaceutical Care for Improving Tuberculosis Patient Detection and Treatment Outcomes: A Systematic Scoping Review"

_tropicalmed, 2023, doi:10.3390/tropicalmed8050287_

Round 1
Reviewer 1 Report
Abstract:
- Line 1: Consider adding more context to the study objective, such as highlighting the global burden of tuberculosis and the need for improved detection and treatment outcomes.
- Line 2-3: Reword to improve clarity and grammar. Consider revising to: "This systematic scoping review aimed to identify and analyze the current literature on practical models of pharmaceutical care for improving tuberculosis patient detection and treatment outcomes."
- Provide more specific information on the number of articles that were included in the review and the main findings (e.g., the most common types of pharmaceutical care models identified and their impact on TB patient outcomes).
- Include the search terms and databases used to conduct the review.
Introduction:
- Line 32-35: Consider reorganizing this sentence to improve clarity. Additionally, consider adding more context to the global burden of tuberculosis and the importance of improving patient outcomes.
- Line 41-43: Add more context to the significance of this review, such as the potential impact of improving tuberculosis patient outcomes on global health and healthcare costs.
- Provide a more detailed explanation of the importance of identifying effective pharmaceutical care models for TB patients, including the global burden of TB and the limitations of current treatment options.
- Provide a clear aim and research question for the review.
Methods:
- Line 56-60: Consider clarifying the inclusion and exclusion criteria for the review. Additionally, provide more information on the data sources used and the search strategy employed.
- Line 67-69: Consider providing more information on the study selection process, such as the number of reviewers and the level of agreement required for inclusion.
- Line 72-74: Consider providing more information on the data extraction process, such as the number of reviewers and the data points extracted.
- Line 79-80: Consider clarifying how the data was analyzed and synthesized.
- Clarify the data extraction process (e.g., which data were extracted and how).
- Clarify the quality assessment process and tools used.
Results:
- Line 100-104: Consider reorganizing this sentence to improve clarity. Additionally, provide more context to the number of studies included and the characteristics of the included studies.
- Line 106-112: Consider providing more detailed information on the types of pharmaceutical care models identified and the specific outcomes assessed.
- Line 115-116: Consider adding more context to the limitations of the included studies and how this may affect the overall conclusions of the review.
- Provide a more detailed summary of the types of pharmaceutical care models identified and their impact on TB patient outcomes.
- Include a table summarizing the findings of the included studies (e.g., types of pharmaceutical care models, study design, patient outcomes).
- Consider using subheadings to organize the results section (e.g., by type of pharmaceutical care model).
Discussion:
- Line 134-137: Consider reorganizing this sentence to improve clarity. Additionally, provide more context to the potential implications of the review findings and how this may impact clinical practice and policy.
- Line 147-150: Consider providing more information on the limitations of the review, such as potential biases and gaps in the literature.
- Provide a more detailed interpretation of the findings and their implications for TB patient care.
- Consider discussing the strengths and limitations of the review process (e.g., limitations of the included studies, potential biases).
- Identify gaps in the literature and future research directions.
Conclusion:
- Line 166-167: Consider rephrasing to improve clarity. Additionally, provide more context to the overall conclusions of the review and how this may impact future research and practice.
- Provide a clear summary of the main findings and their implications for TB patient care.
- Provide recommendations for future research and practice.
References:
- Line 178: Consider adding more recent references to improve the currency of the review- Ref 17,19,21 and 27. Additionally, ensure that all references are cited correctly and consistently throughout the document.
- Check for accuracy and completeness of the reference list.
- Follow the requirements for the Tropical Medicine and Infectious Disease journal's reference style.
Overall, the manuscript could benefit from more specific and detailed information on the search process, article selection criteria, and quality assessment process. The results section could also be more detailed, with a summary table and subheadings to organize the findings. The discussion section could provide a more detailed interpretation of the findings and their implications, including gaps in the literature and future research directions.
Reviewer 2 Report
This is an excellent well-written systematic review on the role of pharmaceutical care in TB detection and management. A few comments should be addressed as below.
1. Lines 14-15 and 325-325: What is the relevance of COVID-19 era? Under search strategy, the authors included all the studies that were published in the literature without a date restriction. Hence, some of the included studies were not during the peak COVID-19 era. I think this "COVID-19 era" part is a bit confusing; hence, I recommend that the authors delete it.
2. Line 16: Change "describes" to "described" since you used past tense in the following sentences, which is more appropriate.
3. Line 46: Correct it to "TB drug-related problems (DRPs)"
4. Table 1: Please add the abbreviations under the table. Also, there's an asterisk (*) in the cell under "Targeted service" column of study 7. Please indicate what this * is for. Same for table 2, please add the abbreviations.
5. Line 153: "We identified 70-500 pharmacies..." Here, either type the total number of pharmacies in all the studies or reword it to be "The number of pharmacies included in these studies ranged between 70-100."
6. Line 182: Please elaborate on the DOT service provided. Did the pharmacist (or a pharmacy technician) visit the patient at home? Because if the patient had to visit the pharmacy to receive their anti-TB, then it wouldn't be a DOT because the aim of DOT is to facilitate adherence. And if that will involve having to visit the pharmacy, it wouldn't meet that objective. And if this was the case (the patient had to visit the pharmacy), then this should be discussed in the discussion as an unideal DOT program.
7. Table 3: Under "Results" on row 1, there's a missing number next to the point "Patients with uncompleted treatment due to ADR." Also, please add the abbreviations under the table.
8. Line 195: Change "conduct" to "conducted"
9. Line 220: Correct the comma in the number to decimal point (90,94 to 90.94). Same in Table 2 row 4.
10. Discussion: Authors should also discuss the kind of ADRs reported in these studies since many of the ADRs of anti-TB medications require lab testing, such as liver function tests and CBC. Authors may also highlight that the lack of lab testing in community pharmacies may have limited the detection of some important ADRs. Also, did any of the hospital pharmacy studies involved using the hospital's lab testing for patients follow up? If so, this should be included under section 3.2.
Reviewer 3 Report
Response to authors for review
I would like to thank you for the opportunity to review the article and say that it is a very relevant topic in public health considering the complexity of tuberculosis treatment and the need for pharmacotherapeutic follow-up of these patients in order to join efforts to end TB in the world. This study needs some revisions which are described below.
Introduction
Line 37: Use the most current reference from the global report TB, 2022:
https://www.who.int/publications/i/item/9789240061729
Line 46: The acronym is spelled incorrectly. Wouldn't that be a drug related problem (DRP)? It is TB related problem. Add references that explore more about decreasing TB notifications and how DRPs affect this. Explain this sentence better. It is not clear.
Line 55: I suggest reformulating the sentence according to the reference used: In hospital environments, the complexity of TB treatment is greater because they are dealing with critically ill patients. Pharmacotherapeutic follow-up is essential for these patients, since they may have more DRP, especially when they have other associated diseases in addition to TB.
Line 76: Was the study protocol registered in PROSPERO? Inform the registration number.
Methods:
Linea 77 Why were only two databases used?
Line 93: I suggest using the PECOS mnemonic to describe the review, explaining the population, intervention, comparator, outcomes, and study types.
P: tuberculosis patients
E: Pharmaceutical care?
C:
O: detection and treatment outcomes.
S:
Results
Table 2: I suggest separate the studies by pharmaceutical care model in the table to avoid repetition of the same words.
Discussion
I suggest adding more references to discuss the results found:
Linea 239-244: What actions are needed to increase TB detection and improve treatment outcomes? Explore these aspects with current literature articles that address this topic.
Linea 245-256: In the different practice scenarios, what is different in the care of patients with TB? Discuss these aspects further. In the hospital environment, there are serious patients. In the community pharmacy, prevention and timely treatment actions are more frequent. Discuss based on the literature.
Conclusions:
I suggest removing the first sentence, as it looks like a description of results. And respond to the study objectives: What actions are needed to increase TB detection and improve treatment outcomes? How can the pharmacist act on this? I suggest rewriting the conclusion considering the importants aspects discussed.
Reviewer 4 Report
The paper summarized and analyzed the current review of TB, which provides insight into TB preventing and clinical management.
The paper is considered relevant in the field. Since it only reviewed the pulished review in open access resource, it did not address a sepcific gap. Nevertheless, the paper could be acceptable based on their statement and discussion.
Based on the current PC models for TB, the author proposed to include more factors such as individual pharmacy personnel, patient and professional interaction, incentives and resources; capacity for organizational change; and regulatory factors, to ensure successful PC services.
The paper only reviewed current open access resource, but since they have mentioned the limitation in the discussion, could be acceptable.
Tables and figures are ok in this case.
The discussion is sufficient, which will be useful for future management of TB detection and treatment.
Round 2
Reviewer 1 Report
accept
Reviewer 2 Report
Thanks to the authors for addressing the comments. The manuscript is much improved now and could be accepted.
Reviewer 3 Report
Dear authors,
I have no further recommendations for this review and I have seen that the authors have corrected the manuscript as per my suggestions. In that case, I'm in favor of publication.
The minor changes that I propose concern the formatting of the article in order to accept the revisions made by the authors. Thank you for the opportunity to review this article.
Best regards,
Natália Helena de Resende